# 🐊 Alligat0R: Pre-Training through Covisibility Segmentation for Relative Camera Pose Regression

**Thibaut Loiseau**[1]  **Guillaume Bourmaud**[2]  **Vincent Lepetit**[1]

[1] LIGM, Ecole des Ponts, Univ. Gustave Eiffel, CNRS, France
[2] Univ. Bordeaux, CNRS, Bordeaux INP, IMS, UMR 5218, France

`{thibaut.loiseau,vincent.lepetit}@enpc.fr`  `guillaume.bourmaud@u-bordeaux.fr`

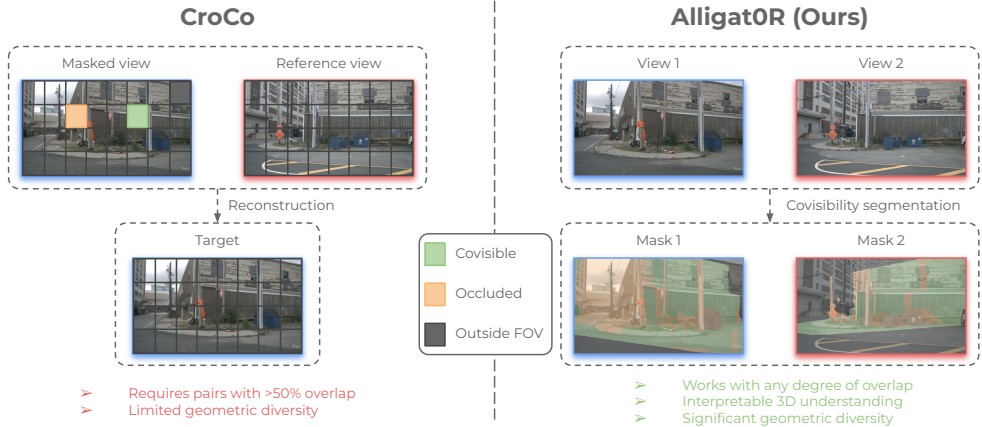

Figure 1: **We introduce Alligat0R, a novel pretraining method for binocular vision.** Alligat0R explicitly segments pixels as covisible, occluded, or outside field-of-view, overcoming the fundamental limitation of CroCo [47, 48] which attempts to reconstruct potentially non-covisible regions.

## Abstract

Pre-training techniques have greatly advanced computer vision, with CroCo's cross-view completion approach yielding impressive results in tasks like 3D reconstruction and pose regression. However, cross-view completion is ill-posed in non-covisible regions, limiting its effectiveness. We introduce Alligat0R, a novel pre-training approach that replaces cross-view learning with a covisibility segmentation task. Our method predicts whether each pixel in one image is covisible in the second image, occluded, or outside the field of view, making the pre-training effective in both covisible and non-covisible regions, and provides interpretable predictions. To support this, we present Cub3, a large-scale dataset with 5M image pairs and dense covisibility annotations derived from the nuScenes and ScanNet datasets. Cub3 includes diverse scenarios with varying degrees of overlap. The experiments show that our novel pre-training method Alligat0R significantly outperforms CroCo in relative pose regression. Alligat0R and Cub3 will be made publicly available.

# 1 Introduction

Pre-training techniques have revolutionized computer vision by enabling large models to learn rich representations [5, 29, 7, 19, 18, 2, 55]. In 3D computer vision, CroCo [47, 48] pioneered cross-view completion as a pretext task, where one view is partially masked and reconstructed using visible portions along with a second reference view of the same scene (see Figure 1 left). This approach allows to learn powerful 3D features and has made possible impressive results on downstream tasks such as 3D reconstruction [45, 14, 49, 42], 3D pose regression [9], camera calibration [26], 4D reconstruction [52, 23, 44] and Gaussian splatting [35, 51].

Despite its success, the cross-view completion pretext task has a fundamental limitation: pretraining is only effective in covisible regions, as the task is ill-posed in non-covisible regions (see Figure 1). As a consequence, Croco [47, 48] relies on pairs with at least 50% of covisible regions.

In this paper, we present Alligat0R, a novel pre-training approach that offers an alternative to the cross-view completion objective through a covisibility segmentation task. Instead of reconstructing masked regions, our method explicitly predicts whether each pixel in one image is: (1) covisible in the second image, (2) occluded, or (3) outside the field of view (FOV). This formulation offers several advantages: it makes the pre-training effective in both covisible and non-covisible regions, it provides interpretable predictions that reveal the model's geometric understanding, and it aligns more directly with the correspondence reasoning required in downstream binocular vision tasks.

To enable our approach, we introduce Cub3, a large-scale dataset comprising two sub datasets, each of 2.5 million image pairs with dense covisibility annotations derived from nuScenes [4] and ScanNet [8] datasets. Cub3 includes challenging scenarios with varying degrees of overlap between views, providing a more diverse training signal than previous approaches.

One might notice that Alligat0R is not strictly self-supervised, unlike some earlier pre-training methods, as it relies on covisibility annotations. However, the same can be said for CroCo, which requires filtering image pairs to retain only those with at least 50% overlap. The creation of our covisibility annotations for nuScenes is more complex than for ScanNet, as the ground truth poses and depth maps are not available, however the process is fully automated. This process is very similar to the one used in RUBIK [28] and relies on registered video sequences and depth predictions.

Our contributions can be summarized as follows:

1. We introduce covisibility segmentation as a novel pre-training objective for binocular vision tasks, replacing the cross-completion-based approach of prior work while maintaining the same network architecture.

2. We create and release Cub3, a large-scale dataset with dense covisibility annotations derived from nuScenes and ScanNet datasets.

3. We show that Alligat0R provides an effective alternative to CroCo pre-training when fine-tuned on the relative pose regression task, particularly for challenging scenarios with limited overlap between views.

4. We demonstrate the effectiveness of our pre-training and provide insights into the model's cross-view reasoning capabilities through interpretable segmentation outputs.

Our experiments demonstrate that explicitly learning to understand covisibility relationships between image pairs leads to more robust and transferable features for relative pose regression compared to reconstruction-based approaches.

# 2 Related Work

**Pretraining on Image Pairs.** To the best of our knowledge, CroCo [47] pioneered the extension of masked image modeling [20] to image pairs, introducing cross-view completion. CroCo v2 [48] applied this cross-view completion framework to large amounts of data. P-Match [56] introduced a variation where both images are partially masked to pre-train an image matching model. Another variant, masked appearance transfer, was proposed in [53] for object tracking, with a similar approach in [36]. Let us highlight that the foundational models DUSt3R [45] and MASt3R [26] are built on CroCo. While all these works rely on cross-view completion, we depart from this framework and propose a novel pretraining objective based on covisibility segmentation.

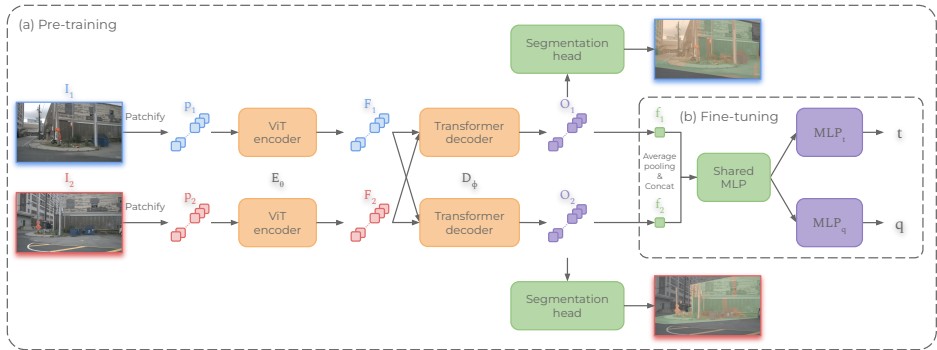

Figure 2: **Overview of Alligat0R.** (a) During pre-training, we use the same architecture as CroCo but replace the reconstruction objective with a covisibility segmentation task, without masking, where each pixel in one view is classified as covisible, occluded, or outside FOV with respect to the other view. (b) For fine-tuning on the relative pose regression task, we pool features from both views, process them through a shared MLP, and use separate heads for predicting translation and rotation.

**Considering covisibility across image pairs.** Ground-truth covisibility masks are widely used in image matching methods [31, 12, 27, 54, 37, 46, 13, 6, 43, 40, 11, 15, 17, 25, 16, 22, 39, 32] to compute overlaps and select training pairs. Several approaches [32, 13, 41, 11, 16, 3] also leverage these masks to learn *matchability* scores, which are used at test time to select correspondences. Additionally, [21] jointly learns covisibility and relative planar pose between $360°$ panoramas. To the best of our knowledge, our work is the first to learn visual representations from covisibility segmentation.

**Fine-tuning for Relative Pose Regression.** Reloc3R [9] recently fine-tuned the foundational model DUSt3R [45], which is based on CroCo, on large-scale data for relative pose regression. In our experiments, we adopt a similar strategy to evaluate our novel pretraining method against CroCo. However, we fine-tune Alligat0R for metric relative translation, whereas Reloc3R predicts only the direction of the relative translation.

## 3 Method

In this section, we describe our novel pre-training approach, Alligat0R, which replaces the cross-view completion objective used in CroCo [47, 48] by a segmentation task. We first outline the overall architecture, which remains largely similar to CroCo, and then detail our covisibility segmentation pre-training objective. Finally, we explain how our model can be fine-tuned for downstream tasks such as relative pose regression.

### 3.1 Architecture Overview

Our architecture closely follows the design of CroCo, which consists of an encoder and a decoder. The encoder is a ViT [10] that processes the input images by dividing them into non-overlapping patches and encoding them into feature representations. The decoder is another transformer that combines information from both views, using cross-attention layers, to make predictions.

We adopt a symmetric architecture for our forward pass, where both images are processed in the same way without any masking. This differs from CroCo, which uses an asymmetric approach that heavily masks one image while leaving the other unmasked. Our symmetric design is not only more efficient but also better aligned with downstream binocular vision tasks, which typically process unmasked images.

As shown in Figure 2 (a), given two images $I_1$ and $I_2$ of the same scene taken from different viewpoints, we first divide both into non-overlapping patches, denoted as tokens $P_1 = \{P_1^1, \ldots, P_1^{N_1}\}$ and $P_2 = \{P_2^1, \ldots, P_2^{N_2}\}$.

The encoder $E_\theta$ processes the tokens from both images independently:

$$\mathbf{F}_1 = E_\theta(\mathbf{P}_1), \quad \mathbf{F}_2 = E_\theta(\mathbf{P}_2), \tag{1}$$

and the decoder $D_\phi$ then takes these features and processes them to enable cross-view reasoning:

$$\mathbf{O}_1 = D_\phi(\mathbf{F}_1, \mathbf{F}_2), \quad \mathbf{O}_2 = D_\phi(\mathbf{F}_2, \mathbf{F}_1), \tag{2}$$

where $\mathbf{O}_1$ and $\mathbf{O}_2$ represent the output features from the decoder. The decoder contains cross-attention mechanisms that allow information exchange between features from both views.

## 3.2 Covisibility Segmentation Pre-Training

A key difference between our approach and CroCo lies in the pre-training objective. Instead of reconstructing masked pixels, we formulate the pre-training task as a covisibility segmentation problem. Our goal is to predict for each pixel in each image whether it is:

1. **Covisible:** the pixel corresponds to a 3D point that is also visible in the other image.
2. **Occluded:** the pixel corresponds to a 3D point that is occluded in the other image.
3. **Outside FOV:** the pixel corresponds to a 3D point that is outside the field of view in the other image.

Formally, for each patch token, the decoder produces a feature vector that is processed by a fully-connected layer to output probabilities for each of the three classes, for each pixel:

$$\hat{\mathbf{Y}}_{ikj} = \mathrm{softmax}(\mathbf{W} \cdot \mathbf{O}_{ik} + \mathbf{b})_j, \tag{3}$$

where $\hat{\mathbf{Y}}_{ikj} \in \mathbb{R}^3$ represents the predicted probabilities for the three covisibility classes for pixel $j$ in patch $k$ of image $i$, and $\mathbf{W}$ and $\mathbf{b}$ are learnable parameters.

During pre-training, we optimize the network using a cross-entropy loss in each image:

$$\mathcal{L}_{\mathrm{ce}_i} = -\frac{1}{N_i} \sum_{j=1}^{N_i} \ln(\hat{\mathbf{Y}}_{ij,c_{ij}}), \tag{4}$$

where $\hat{\mathbf{Y}}_{ij,c_{ij}}$ is the predicted probability for the ground-truth class $c_{ij}$ in pixel $j$ of image $i$, and $N_i$ is the number of pixels in each image. The total cross-entropy loss is the sum over both images: $\mathcal{L}_{\mathrm{ce}} = \mathcal{L}_{\mathrm{ce}_1} + \mathcal{L}_{\mathrm{ce}_2}$.

This formulation offers several advantages over the cross-view-completion-based approach. First, it makes the pre-training effective in both covisible and non-covisible regions, as we explicitly model cases where pixels have no covisible region in the other view. Second, it provides interpretable outputs that directly reveal the model's understanding of scene geometry. Third, it better aligns with downstream tasks such as pose regression, where both images are fully visible.

## 3.3 Fine-Tuning for Relative Pose Regression

After pre-training, we fine-tune our model for the task of relative pose regression. We add a pose regression head on top of the pre-trained encoder-decoder architecture, while keeping the original covisibility segmentation head. This design allows the model to leverage the geometric understanding acquired during pre-training.

The pose regression head consists of several components as shown in Figure 2 (b). First, we apply global average pooling to the decoder outputs for both images:

$$\mathbf{f}_1 = \mathrm{GlobalAvgPool}(\mathbf{O}_1), \quad \mathbf{f}_2 = \mathrm{GlobalAvgPool}(\mathbf{O}_2). \tag{5}$$

The pooled features are concatenated and processed through a shared MLP:

$$\mathbf{f}_{\mathrm{shared}} = \mathrm{MLP}([\mathbf{f}_1, \mathbf{f}_2]). \tag{6}$$

We then use separate heads for predicting the metric relative translation vector $\hat{\mathbf{t}} \in \mathbb{R}^3$ and the relative rotation represented as a quaternion $\hat{\mathbf{q}} \in \mathbb{R}^4$:

$$\hat{\mathbf{t}} = \mathrm{MLP}_{\mathbf{t}}(\mathbf{f}_{\mathrm{shared}}), \quad \hat{\mathbf{q}} = \mathrm{MLP}_{\mathbf{q}}(\mathbf{f}_{\mathrm{shared}}). \tag{7}$$

We use unit quaternions with L2 normalization, treating them as 4D vectors in the homoscedastic loss function.

Our fine-tuning stage consists of two phases:

1. First, we freeze the pre-trained encoder, decoder and covisibility segmentation head, and only train the pose regression head. During this phase, we use a homoscedastic loss [24] combining MSE losses for translation and quaternion predictions:

$$\mathcal{L}_{\text{pose}} = \frac{1}{2\sigma_{\mathbf{t}}^2} \|\mathbf{t} - \hat{\mathbf{t}}\|^2 + \frac{1}{2\sigma_{\mathbf{q}}^2} \|\mathbf{q} - \hat{\mathbf{q}}\|^2 + \log \sigma_{\mathbf{t}} + \log \sigma_{\mathbf{q}} \,, \tag{8}$$

where $\mathbf{t}$ and $\mathbf{q}$ are the ground-truth metric translation and quaternion respectively, and $\sigma_{\mathbf{t}}$ and $\sigma_{\mathbf{q}}$ are learnable parameters that automatically balance the two loss terms.

2. In the second phase, we unfreeze the backbone and the covisibility segmentation head, and train the full network with a joint loss that combines the pose regression loss and the covisibility segmentation loss:

$$\mathcal{L}_{\text{joint}} = \mathcal{L}_{\text{pose}} + \frac{1}{2\sigma_{\text{seg}}^2} \mathcal{L}_{\text{ce}} + \log \sigma_{\text{seg}} \,, \tag{9}$$

where $\sigma_{\text{seg}}$ is a learnable parameter.

This training strategy ensures that the model maintains its interpretable covisibility segmentation capabilities while being optimized for the pose regression task.

# 4 Cub3: A Large-Scale Covisibility Dataset

To enable our covisibility segmentation approach, we introduce Cub3, a large-scale dataset comprising two sub datasets each of 2.5 million image pairs with dense covisibility annotations derived from both the autonomous driving nuScenes dataset [4], and the indoor ScanNet [8] dataset. Cub3 will be made publicly available.

## 4.1 Dataset Construction

**nuScenes –** We leverage the covisibility estimation pipeline introduced in RUBIK [28] to generate pixel-level covisibility annotations for image pairs from nuScenes. In brief, this pipeline uses monocular metric depth predictions from UniDepth [30] and surface normals from Depth Anything V2 [50], combined with camera poses from COLMAP [33] reconstructions, to automatically classify each pixel as either covisible, occluded, or outside FOV with respect to another image. We apply this pipeline to all possible image pairs within each scene, resulting in approximately 34 million annotated pairs.

**ScanNet –** The pipeline is similar but simpler for ScanNet, as we rely on the ground truth depths and camera poses to get all covisibility annotations.

For our experiments, we created two dataset variants, each containing 5M image pairs (2.5M from nuScenes and 2.5M from ScanNet):

• **Cub3-50:** Image pairs with at least 50% overlap, similar to the criterion used in CroCo [48]. This dataset provides a direct comparison point with existing cross-view completion approaches.

• **Cub3-all:** Image pairs with at least 5% overlap. This dataset, contains challenging image pairs to test the robustness of methods to handle low-overlap scenarios that are common in real-world applications.

Figure 3 shows examples or our covisibility annotations on image pairs from Cub3, illustrating how our process classifies pixels into the three categories across varying levels of difficulty.

It is also worth noting that our annotation process, especially on nuScenes, inherits some limitations from the underlying monocular depth estimation models. The depth predictions may occasionally struggle with challenging scenarios such as reflective surfaces, transparent objects, or regions with complex geometry or far-away objects. Consequently, some annotations in our dataset, particularly the distinction between covisible and occluded pixels, may contain noise. Despite these imperfections, our experiments show that the scale and diversity of Cub3 enable effective training of robust covisibility segmentation models.

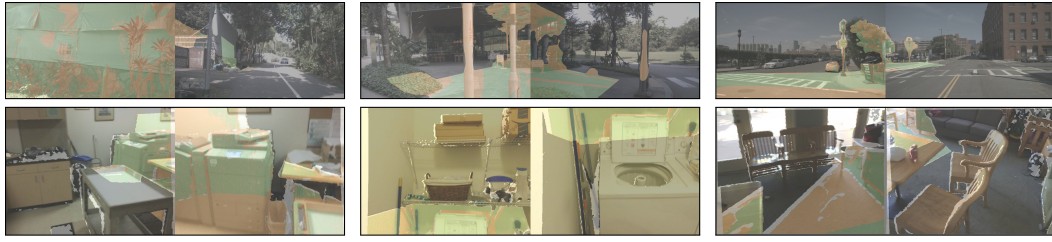

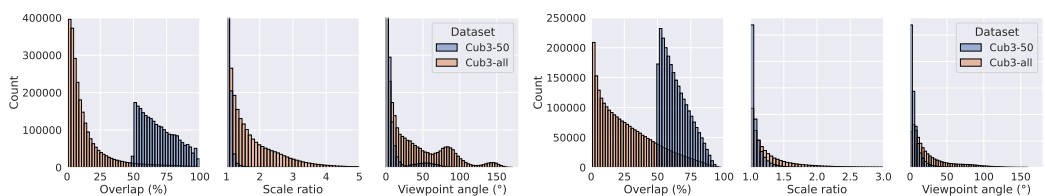

Figure 3: Covisibility annotation examples from Cub3 for nuScenes (top) and ScanNet (bottom). For each image pair, we show the corresponding covisibility maps with color-coding for covisible, occluded, and outside FOV regions. Note how our annotation process handles varying degrees of overlap and challenging viewpoint changes. Let us highlight that some annotations, particularly the distinction between covisible and occluded pixels, may contain noise, especially for nuScenes, and we demonstrate in the experiments that Alligat0R is highly robust to this noise.

Figure 4: Distributions of overlap, scale ratio, and viewpoint angle in Cub3-all and Cub3-50 for nuScenes (left) and ScanNet (right).

## 4.2 Dataset Statistics

Figure 4 illustrates the distribution of our datasets across three geometric criteria introduced in RUBIK [28]: overlap percentage, scale ratio, and viewpoint angle. The Cub3-all dataset's distribution is heavily skewed towards very challenging pairs, which are particularly difficult for cross-completion methods like CroCo. In contrast, our covisibility segmentation approach is designed to benefit from such challenging cases effectively.

Given our computational resources, Cub3 is currently limited to either autonomous driving scenarios from nuScenes, or indoor scenes from ScanNet. But such scenarios have very important applications, and given its size, variety, and challenges, Cub3 is already sufficient to validate Alligat0R—very much like the initial version of CroCo [47] was demonstrated as a "proof-of-concept":

- **Scale:** With 5M annotated pairs, Cub3 provides sufficient data to train and validate our approach on both outdoor and indoor scenarios.
- **Geometric variety:** It covers a wide range of overlaps, scales and viewpoint angles within the driving and indoor contexts.
- **Challenging scenarios:** It includes pairs with minimal overlap that test the limits of CroCo, as was done in the original paper.

Our experiments on the RUBIK benchmark, which was specifically designed to evaluate challenging autonomous driving scenarios, as well as results on ScanNet1500 test set demonstrate the effectiveness of our approach. The promising results we obtain motivate future work to extend Cub3 to more diverse environments beyond urban driving scenes and indoor scenes. However, even within this specific domain, our method shows significant improvements over CroCo, particularly in geometrically challenging configurations.

# 5  Experiments

## 5.1  Implementation Details

We implement Alligat0R using PyTorch and conduct all experiments on NVIDIA A100 GPUs. The model architecture follows the design described in Section 3.1. The input images are classically resized to have a maximum dimension of 512 pixels while maintaining their aspect ratio. More information is provided in the supplementary material.

## 5.2  Main Results

Table 1: Results on RUBIK [28] and ScanNet1500 [8] for metric relative pose regression. For all experiments, fine-tuning is performed using Cub3-all.

| Pre-training | Pre-Training Set | Backbone | RUBIK | | | ScanNet1500 | | |
|---|---|---|---|---|---|---|---|---|
| | | | 5° / 0.5m | 5° / 2m | 10° / 5m | 10° / 0.25m | 10° / 0.5m | 10° / 1m |
| CroCo | Cub3-50 | ❄ | 4.4 | 21.8 | 48.1 | 48.3 | 64.1 | 69.3 |
| CroCo | Cub3-all | ❄ | 2.4 | 8.9 | 25.2 | 8.7 | 18.3 | 24.9 |
| Alligat0R | Cub3-50 | ❄ | 9.3 | 32.4 | 58.4 | 47.5 | 59.9 | 63.4 |
| Alligat0R | Cub3-all | ❄ | 24.0 | 55.3 | **82.3** | 78.2 | 90.1 | 92.5 |
| CroCo | Cub3-50 | 🔥 | 12.4 | 38.3 | 66.7 | 75.7 | 87.4 | 91.5 |
| CroCo | Cub3-all | 🔥 | 12.5 | 38.6 | 66.2 | 64.1 | 79.1 | 85.0 |
| Alligat0R | Cub3-50 | 🔥 | 21.2 | 53.3 | 78.1 | 82.8 | 91.7 | 93.9 |
| Alligat0R | Cub3-all | 🔥 | **24.6** | **60.3** | 81.9 | **85.5** | **92.5** | **95.1** |
| From scratch | Cub3-all | 🔥 | 10.8 | 29.1 | 52.3 | 37.0 | 51.8 | 57.2 |

**Metric Relative Pose Regression Performance.** We evaluate our proposed Alligat0R pre-training approach on the metric relative pose regression task and compare it with CroCo pre-training using the same architecture and fine-tuning protocol. Table 1 presents the results on the RUBIK [28] benchmark, which is specifically designed to evaluate performance across different geometric challenges in autonomous driving scenarios, and on the indoor ScanNet1500 [8] benchmark.

As shown in Table 1, Alligat0R consistently outperforms CroCo across different training and evaluation configurations. Notably, when pre-trained on Cub3-all and fine-tuned with the backbone frozen, Alligat0R achieves significantly higher performance and reaches state-of-the-art on the RUBIK benchmark (55.3% success rate at $5°$/2m and 82.3% at $10°$/5m) compared to the CroCo model pre-trained on the same dataset (which only achieves 8.9% and 25.2% respectively in the frozen backbone configuration). This substantial improvement demonstrates the effectiveness of our covisibility segmentation pre-training approach, especially for challenging scenarios with varying degrees of overlap.

**Impact of Training Data Distribution.** A key observation is the contrasting behavior of Alligat0R and CroCo when trained on different data distributions. While CroCo performs similarly or better when trained on easier pairs (Cub3-50) compared to harder pairs (Cub3-all), Alligat0R shows a different trend. Our method systematically improves when trained on the full distribution (Cub3-all). This demonstrates that Alligat0R can effectively learn from challenging examples, while CroCo struggles to benefit from them.

Figure 5 further illustrates this advantage by breaking down performance across different overlap percentages, scale ratios and viewpoint angles. While both methods perform well on high-overlap pairs (80-100%), Alligat0R trained on Cub3-all maintains strong performance even as overlap decreases, whereas CroCo's accuracy drops dramatically for pairs with less than 40% overlap. This demonstrates Alligat0R's ability to generalize across a wide range of geometric configurations, which is crucial for real-world applications where overlap cannot be guaranteed.

**Fine-tuning Strategy Comparison.** We also compare different fine-tuning strategies, specifically the impact of freezing versus unfreezing the backbone during fine-tuning. One can see that Alligat0R trained on Cub3-all is already quite high with a frozen backbone and always outperforms CroCo even when its backbone is unfrozen. This suggests that our pre-training strategy is better aligned with the downstream pose regression task.

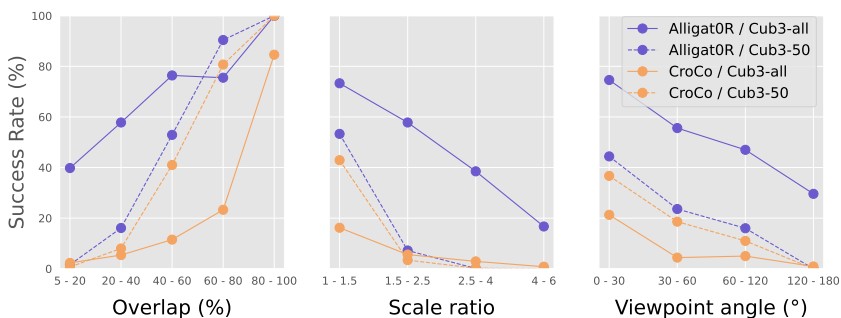

Figure 5: Performance of CroCo and Alligat0R across different geometric challenges on RUBIK. Results show accuracy at the $5°/2m$ threshold for models trained on different datasets (Cub3-50 or Cub3-all) and for frozen backbones. Alligat0R trained on Cub3-all consistently outperforms other configurations, particularly for challenging cases with low overlap, large scale differences, and extreme viewpoint changes.

**Comparison with Training from Scratch.** To isolate the impact of our pre-training approach, we compare the performance of a model trained from scratch for relative pose regression versus those initialized with Alligat0R or CroCo pre-training. The bottom row of Table 1 shows that training from scratch achieves notably lower performance, highlighting the value of the representations learned during pre-training.

**Robustness to Label Noise.** To address concerns about sensitivity to covisibility segmentation errors, we conducted a sensitivity analysis by training Alligat0R on ScanNet with 20% random label noise injected during pre-training. Surprisingly, this actually improved performance across all thresholds (+2.2%, +2.4%, +0.8% respectively), as shown in Table 2. This counterintuitive result suggests that label noise acts as a form of regularization similar to label smoothing [38], encouraging learning of more generalizable features. Importantly, this demonstrates that our approach remains robust even when covisibility annotations are imperfect.

Table 2: Sensitivity analysis: Impact of label noise on Alligat0R performance on ScanNet1500. Results show that 20% random label noise during pre-training actually improves performance, demonstrating robustness to annotation errors.

| Method | Noise | ScanNet1500 $10°$ / $0.25m$ | $10°$ / $0.5m$ | $10°$ / $1m$ |
|---|---|---|---|---|
| Alligat0R | 0% | 85.5 | 92.5 | 95.1 |
| Alligat0R | 20% | **87.7** | **94.9** | **95.9** |
| CroCo | N/A | 75.7 | 87.4 | 91.5 |

Table 3: **RUBIK Benchmark.** The results are reported from [28]. Success rate (in %) for each method across individual geometric criterion bins. Best and second-best values for each column are shown in **bold** and underlined respectively.

| | Overlap (%) | | | | | Scale Ratio | | | | Viewpoint Angle (°) | | | | Whole | Time |
|---|---|---|---|---|---|---|---|---|---|---|---|---|---|---|---|
| | 80–100 | 60–80 | 40–60 | 20–40 | 5–20 | 1.0–1.5 | 1.5–2.5 | 2.5–4.0 | 4.0–6.0 | 0–30 | 30–60 | 60–120 | 120–180 | Dataset | (ms) |
| *Detector-free methods* | | | | | | | | | | | | | | | |
| LoFTR [37] | 87.2 | 88.4 | 47.2 | 17.5 | 5.0 | 51.6 | 10.1 | 2.3 | 0.6 | 43.2 | 27.9 | 15.1 | 0.0 | 24.9 | 185 |
| ELoFTR [46] | 56.4 | 90.3 | 50.8 | 22.1 | 6.3 | 51.2 | 15.6 | 4.4 | 0.7 | 42.2 | 30.8 | 18.2 | 0.1 | 26.6 | 124 |
| ASpanFormer [6] | 72.2 | 72.3 | 44.5 | 21.9 | 7.4 | 46.0 | 14.9 | 6.9 | 1.6 | 42.5 | 27.2 | 16.0 | 0.1 | 24.8 | 108 |
| RoMa [13] | 67.0 | **98.3** | 84.5 | 52.7 | 20.2 | 71.2 | 43.2 | 26.6 | 8.3 | 57.5 | 56.2 | 44.1 | 3.0 | 47.3 | 614 |
| DUSt3R [45] | 81.8 | 97.4 | **90.8** | 58.4 | 30.4 | 73.3 | 57.9 | 40.1 | 9.9 | 67.4 | 55.3 | 50.0 | 35.2 | 54.8 | 257 |
| MASt3R [26] | 52.0 | 97.5 | 89.6 | 61.0 | 28.4 | 71.2 | 52.3 | 42.5 | 13.8 | 53.5 | **65.6** | **54.5** | 14.1 | 53.6 | 173 |
| *Relative pose regression* | | | | | | | | | | | | | | | |
| **Alligat0R (Ours)** | **100.0** | 89.7 | 80.2 | **61.5** | **44.3** | **78.4** | **61.3** | **43.9** | **23.2** | **77.5** | 62.7 | 51.3 | **37.3** | **60.3** | **57** |

| Reference | Target | Masked | CroCo recons. | Alligat0R segmentations |
|---|---|---|---|---|

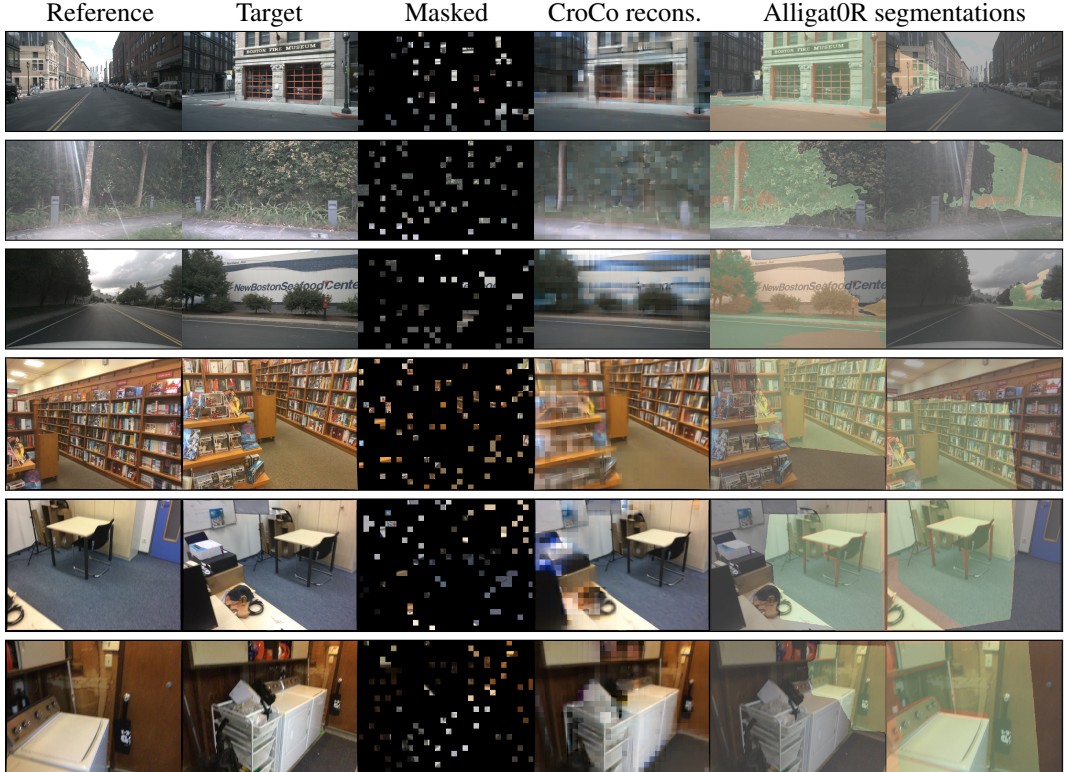

Figure 6: **Qualitative comparison.** CroCo's reconstructions are blurred in masked non-covisible regions, while Alligat0R often correctly identifies covisible, occluded, outside FOV regions across varying degrees of overlap and viewpoint changes.

## 5.3 Qualitative Analysis

**Covisibility and Reconstruction Visualization.** Figure 6 shows qualitative examples of the covisibility segmentation and CroCo reconstructions produced by our models on pairs from Cub3. Alligat0R successfully identifies covisible regions, occluded areas, and regions outside FOV across varying degrees of overlap and viewpoint changes. These visualizations provide insights into the geometric understanding acquired by our model. Conversely, while Croco succeeds in reconstructing masked covisible regions, the reconstructions are blurred in masked non-covisible regions as cross-completion is ill-posed in such areas.

## 5.4 Comparison with state-of-the-art on RUBIK

In Table 3, we evaluate the performance of Alligat0R fine-tuned for metric relative pose estimation against state-of-the-art methods on RUBIK [28]. Let us highlight that:

- Alligat0R was trained on nuScenes (on scenes that are not part of RUBIK),
- whereas none of the state-of-the-art methods used nuScenes as training set.

This gives an advantage to Alligat0R over the other methods. However, RUBIK allows the evaluation over large ranges of three geometric criteria—overlap, scale ratio and viewpoint angle—which we found interesting. Despite this advantage, we argue that Alligat0R's performance remains remarkable, as pretraining with CroCo, in contrast, leads to a significant performance drop (see Table 1).

With an overall success rate of 60.3%, Alligat0R ranks first. Notably, it significantly outperforms all other methods on the hardest bin across the three geometric criteria, which confirms the advantage of its pretraining on difficult image pairs. While Alligat0R's training set was highly skewed toward challenging pairs (see Figure 4), one might expect a performance drop on easier bins. However, this does not happen: for instance, Alligat0R still ranks first for small scale ratios and small viewpoint angles. Finally, Alligat0R is among the fastest methods, as it directly regresses the relative pose rather than producing intermediate correspondences.

## 5.5 Out-of-Domain Generalization

To demonstrate the generalization capabilities of our method beyond the training domains, we evaluate Alligat0R on zero-shot correspondence estimation using the ETH3D dataset [34]. We use the correlations from decoder features to estimate correspondences and measure performance using Average EndPoint Error (AEPE) as described in [1].

**Zero-shot Correspondence Estimation.** As detailed in the supplementary material, Alligat0R consistently outperforms CroCo when trained on the same datasets, and even surpasses CroCo v2 which was pre-trained on 5 datasets. This demonstrates that our covisibility segmentation pre-training learns more generalizable features for dense correspondence tasks.

**Comparison with Official CroCo v2.** We also compare against the official CroCo v2 model, fine-tuned on our Cub3-all dataset for pose regression. As shown in Table 4, Alligat0R significantly outperforms CroCo v2 across all metrics, demonstrating the effectiveness of our covisibility segmentation approach even when compared to a model pre-trained on multiple datasets.

Table 4: Comparison with official CroCo v2 fine-tuned on Cub3-all dataset for pose regression on RUBIK benchmark. Alligat0R significantly outperforms CroCo v2 across all metrics.

| Method | RUBIK | | |
| --- | --- | --- | --- |
| | $5° / 0.5m$ | $5° / 2m$ | $10° / 5m$ |
| Alligat0R (Frozen) | **24.0** | **55.3** | **82.3** |
| CroCo v2 (Frozen) | 5.8 | 15.8 | 34.1 |
| Alligat0R (Unfrozen) | **24.6** | **60.3** | **81.9** |
| CroCo v2 (Unfrozen) | 14.6 | 44.5 | 73.1 |

# 6 Limitations

While our approach demonstrates significant improvements over CroCo, it has several limitations:

- First, our current evaluation is limited to urban driving scenes from nuScenes and indoor scenes from ScanNet. The performance of Alligat0R on more diverse environments, such as natural scenes or extreme weather conditions, remains to be investigated.

- Second, our pre-training method requires ground truth depth maps and poses to produce dense covisibility annotations. While we investigated a fully automated pseudo ground truth generation for nuScenes, such a pre-processing is computationally intensive. This dependency on 3D data might limit the scalability of our approach to scenarios where such data is not readily available.

- Finally, while we show promising results on metric relative pose regression, the effectiveness of our pre-training approach for other binocular vision tasks, such as stereo matching or optical flow estimation, remains to be explored.

# 7 Conclusion and Future Work

We introduced Alligat0R, a novel pre-training approach that replaces cross-view completion with covisibility segmentation. Unlike CroCo's masked reconstruction, our approach explicitly models covisibility relationships, enabling more effective learning from challenging geometric configurations.

We demonstrated significant improvements over CroCo on metric relative pose regression and zero-shot correspondence estimation, particularly on challenging scenes with limited overlap. Our approach's robustness to geometrically challenging examples and superior generalization capabilities highlight the effectiveness of covisibility segmentation for binocular vision tasks.

We created Cub3, a large-scale dataset of 5M image pairs with dense covisibility annotations from nuScenes and ScanNet. Our work demonstrates that explicitly modeling covisibility provides an effective alternative to binocular vision tasks compared to reconstruction-based methods.

## Acknowledgments

This project has received funding from the Bosch Research Foundation (Bosch Forschungsstiftung), the European Union (ERC Advanced Grant explorer Funding ID #101097259), and was granted access to the HPC resources of IDRIS under the allocation 2024-AD011014905R1 made by GENCI.

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

# A Supplementary Material

This supplementary material provides additional details about our Alligat0R model, including training curves, implementation details, and visualizations that complement the main paper.

## A.1 Pre-training Learning Curves

Figure 7 shows the learning curves for the pre-training phase of both CroCo and Alligat0R on the Cub3-50 and Cub3-all datasets. Alligat0R's loss converges smoothly on both datasets, indicating stable training.

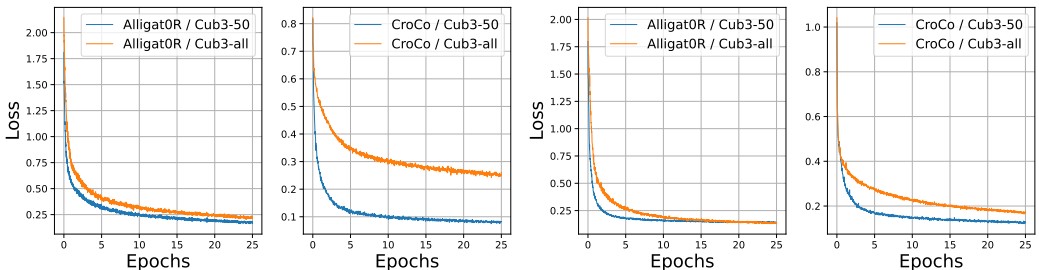

Figure 7: Learning curves during pre-training for CroCo (reconstruction loss) and Alligat0R (segmentation loss) on Cub3-50 and Cub3-all datasets for nuScenes (left) and ScanNet (right). Alligat0R shows stable convergence on both datasets.

## A.2 Fine-tuning Learning Curves

Figure 8 presents the fine-tuning curves for the pose regression task. The plots show the full loss during training (Eq. 9 in the main paper for Alligat0R, Eq. 8 for CroCo). Alligat0R fine-tuned on Cub3-all shows faster convergence and reaches higher accuracy than the same configuration for CroCo, highlighting the transferability of features learned through covisibility segmentation on difficult pairs. The loss increase at 5 epochs corresponds to unfreezing the backbone. For Alligat0R, at 5 epochs we also add the segmentation loss to maintain interpretability.

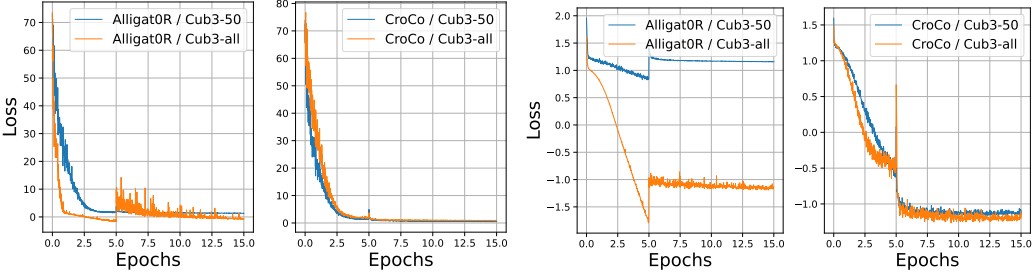

Figure 8: Learning curves during fine-tuning for pose regression for nuScenes (left) and ScanNet (right). Alligat0R pre-trained on Cub3-all converges faster and achieves higher success rates than other methods, demonstrating the effectiveness of our covisibility segmentation pre-training approach.

## A.3 Detailed Performance Analysis

Figure 9 provides a comprehensive breakdown of performance across different geometric criteria on RUBIK. This visualization emphasizes Alligat0R's strong performance across all difficulty ranges, particularly in challenging scenarios where traditional methods struggle.

## A.4 Implementation Details

We use a ViT-based encoder and transformer decoder backbone similar to CroCo, with 24 layers for the encoder and 12 for the decoder. For pre-training, we use the AdamW optimizer with a learning

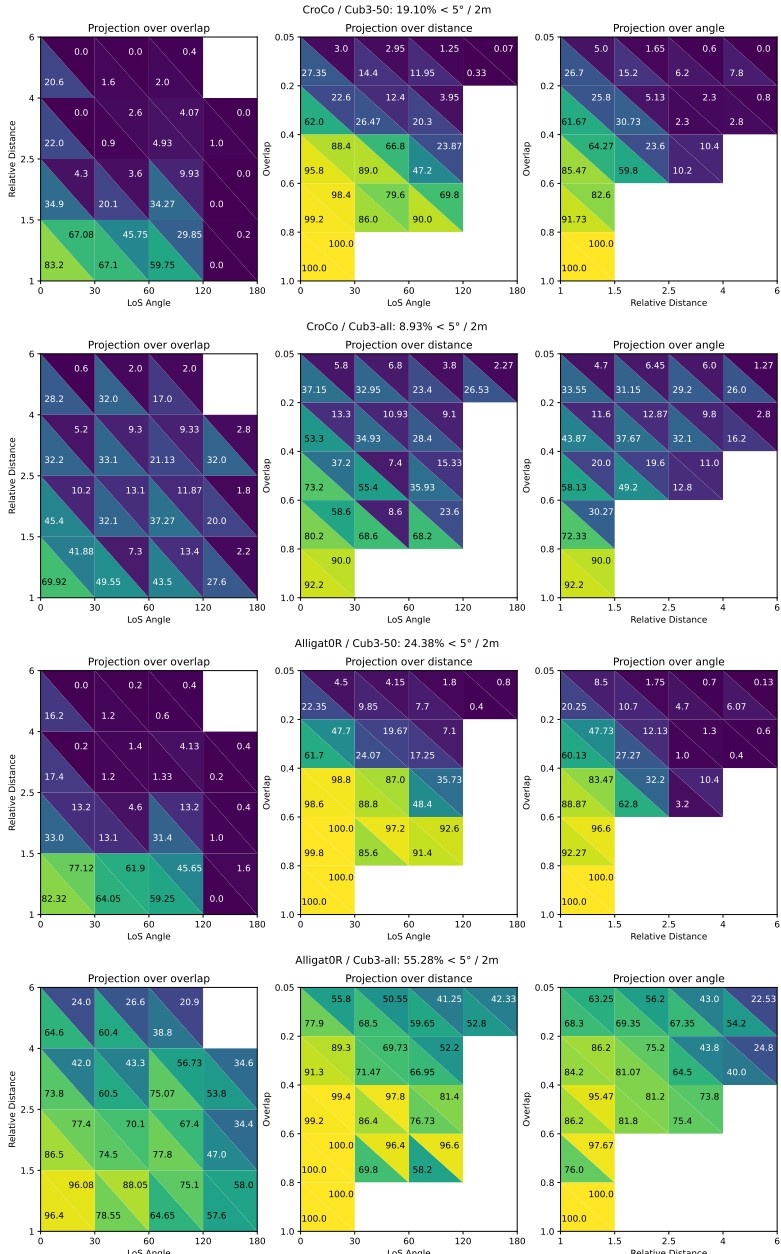

Figure 9: **Detailed breakdown of performance across different geometric criteria.** Success rate either for R@5° or t@2m (bottom-left and top-right of each triangle, respectively), when projecting results onto individual geometric criteria of RUBIK. For each method, we show three plots corresponding to the projection over overlap (left), scale ratio (middle), and viewpoint angle (right).

rate of 1.5e-4, weight decay of 0.05, and a batch size of 32 per GPU. We employ a cosine learning rate schedule with 2 epochs of warmup and train for 25 epochs on our Cub3 datasets.

For fine-tuning on relative pose regression, we follow the two-phase approach described in Section 3.3 of the main paper:

- Phase 1: Freeze the backbone and train only the pose regression head for 5 epochs with a learning rate of 1e-4

- Phase 2: Unfreeze the entire network and jointly train with both the pose regression loss and covisibility segmentation loss for an additional 10 epochs with a learning rate of 5e-5

The comprehensive architecture of Alligat0R is illustrated in Figure 2 of the main paper, showing both the pre-training and fine-tuning phases. During pre-training, the model learns to segment pixels in each view as covisible, occluded, or outside the field of view with respect to the other view. During fine-tuning, we introduce a pose regression head while maintaining the segmentation capability to leverage the geometric understanding acquired during pre-training.

### A.5   Data Splits and Sampling Protocol

To ensure proper evaluation and avoid data leakage, we carefully separate training and test data:

**Data Splits:**

- **RUBIK benchmark:** Uses test scenes from nuScenes, while our Cub3 dataset uses only the training split of nuScenes, with completely disjoint scenes.
- **ScanNet1500 benchmark:** Derived from the standard ScanNet test split, whereas Cub3 uses scenes from the ScanNet training split exclusively.

**Sampling Protocol:** We pre-filter all possible pairs from all training scenes with at least 5% overlap for the "all" version of Cub3, and at least 50% for the "50" version. We then sample from all those pre-filtered pairs to extract the desired number of samples and covisibility masks. The RUBIK benchmark creation follows the protocol described in [28] and is extracted from test scenes, while ScanNet1500 was created from test scenes of ScanNet as first introduced in [32].

### A.6   Additional Ablation Studies

#### A.6.1   Impact of the Number of Classes

We implemented two variants of Alligat0R to investigate the importance of our pre-training with the three classes (covisible, occluded, outside FOV). We tried pre-training with only two classes, either covisible or not (in this case occluded and outside FOV are merged), or inside FOV or not (in this case, covisible and occluded are merged) on Cub3-all for nuScenes. The results on the RUBIK benchmark are presented in Table 5.

Table 5: Results on RUBIK for **metric** relative pose regression when pre-training with only two classes. For all experiments, pre-training and fine-tuning is performed using Cub3-all.

| | RUBIK | | |
|---|---|---|---|
| Classes | 5° / 0.5m | 5° / 2m | 10° / 5m |
| Covisible or not | 22.5 | 55.7 | 80.0 |
| Inside FOV or not | **24.6** | 59.6 | **81.9** |
| All 3 classes | **24.6** | **60.3** | **81.9** |

While the performance improvement with three classes is modest, we believe that the model's knowledge about occluded regions could be beneficial for other tasks. As noted in the main paper, the annotated maps may contain noise between covisible and occluded zones due to reliance on monocular depth predictions.

#### A.6.2   Non-metric Relative Pose Regression

We investigated whether our pre-training is useful for non-metric relative pose regression (regressing only the angle for translation) and compared our results with CroCo pre-training on the ScanNet1500 benchmark. For this experiment, we changed our pose regression head using the one from Reloc3r, along with its loss function. The results are shown in Table 6.

Alligat0R significantly outperforms CroCo on this task, demonstrating the versatility of our pre-training approach.

Table 6: Results on ScanNet1500 for **non-metric** relative pose regression. For Alligat0R, pre-training and fine-tuning is performed using Cub3-all, whereas for CroCo, pre-training is performed using Cub3-50 and fine-tuning using Cub3-all.

| | ScanNet1500 | | |
|---|---|---|---|
| Pre-Training | AUC@5 | AUC@10 | AUC@20 |
| CroCo | 13.2 | 34.1 | 57.1 |
| Alligat0R | **20.5** | **43.9** | **66.2** |

## A.7    Zero-shot Correspondence Estimation on ETH3D

To demonstrate the generalization capabilities of our method beyond the training domains, we evaluate Alligat0R on zero-shot correspondence estimation using the ETH3D dataset [34]. We use the correlations from decoder features to estimate correspondences and measure performance using Average EndPoint Error (AEPE) as described in [1].

Table 7: Zero-shot correspondence estimation on ETH3D dataset using AEPE. Alligat0R consistently outperforms CroCo variants, demonstrating better generalization to out-of-domain dense matching tasks.

| Method | Training Data | AEPE ($\downarrow$) |
|---|---|---|
| Alligat0R | nuScenes-all | **43.82** |
| CroCo | nuScenes-all | 77.61 |
| Alligat0R | nuScenes-50 | **44.12** |
| CroCo | nuScenes-50 | 92.98 |
| Alligat0R | ScanNet-all | **36.07** |
| CroCo | ScanNet-all | 56.70 |
| Alligat0R | ScanNet-50 | **38.45** |
| CroCo | ScanNet-50 | 86.60 |
| CroCo v2 | 5 datasets | 51.55 |

The results in Table 7 show that Alligat0R consistently outperforms CroCo when trained on the same datasets, and even surpasses CroCo v2 which was pre-trained on 5 datasets. This demonstrates that our covisibility segmentation pre-training learns more generalizable features for dense correspondence tasks.

## A.8    Additional Qualitative Results

We provide additional visualizations from our nuScenes and ScanNet datasets in Figures 10 and 11, along with more predictions from Alligat0R and CroCo in Figure 12.

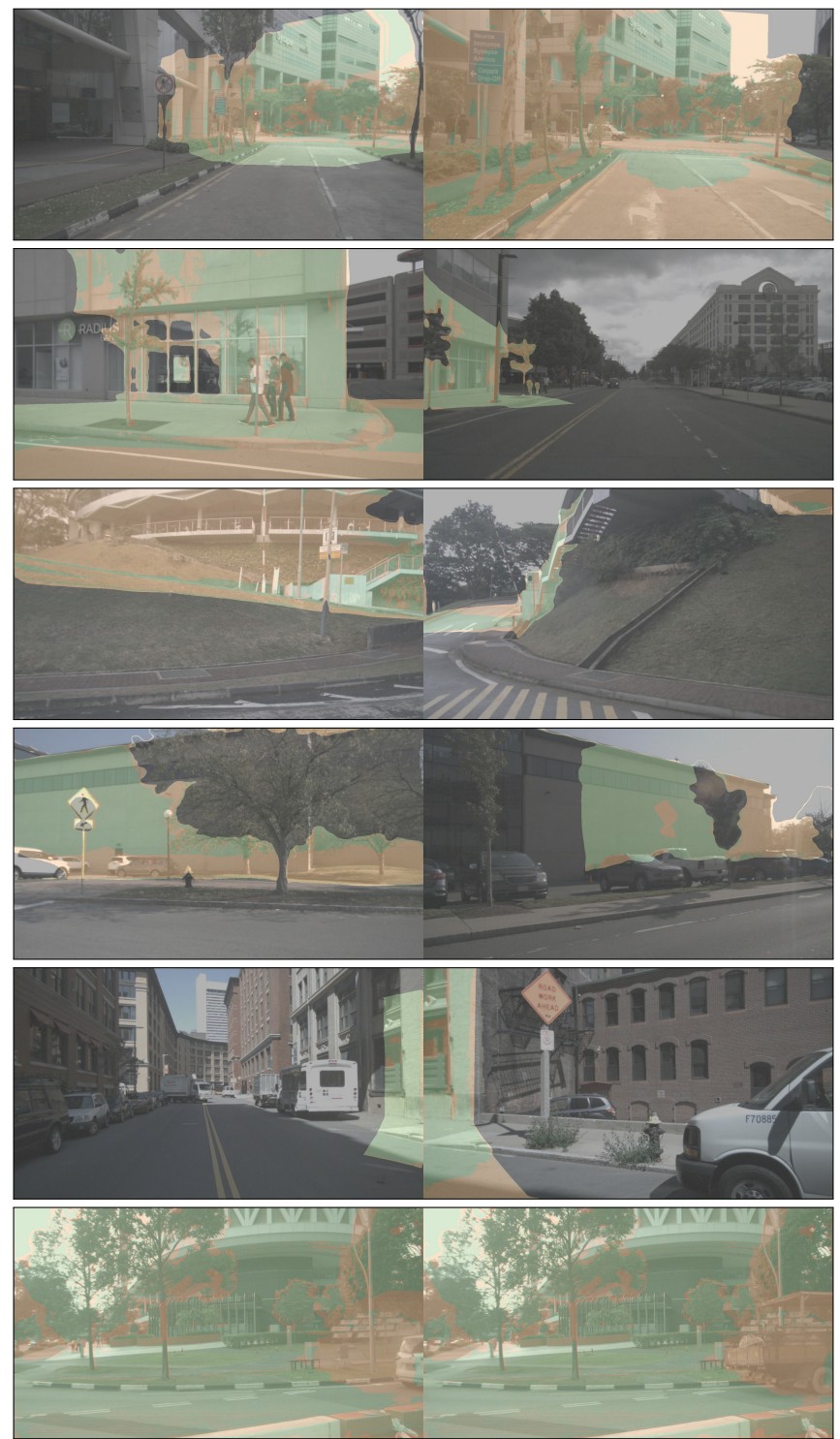

Figure 10: Covisibility annotation examples from Cub3 for nuScenes. For each image pair, we show the corresponding covisibility maps with color-coding for covisible, occluded, and outside FOV regions. Note how our annotation process handles varying degrees of overlap and challenging viewpoint changes. Let us highlight that some annotations, particularly the distinction between covisible and occluded pixels, may contain noise, especially for nuScenes, and we demonstrate in the experiments that Alligat0R is highly robust to this noise.

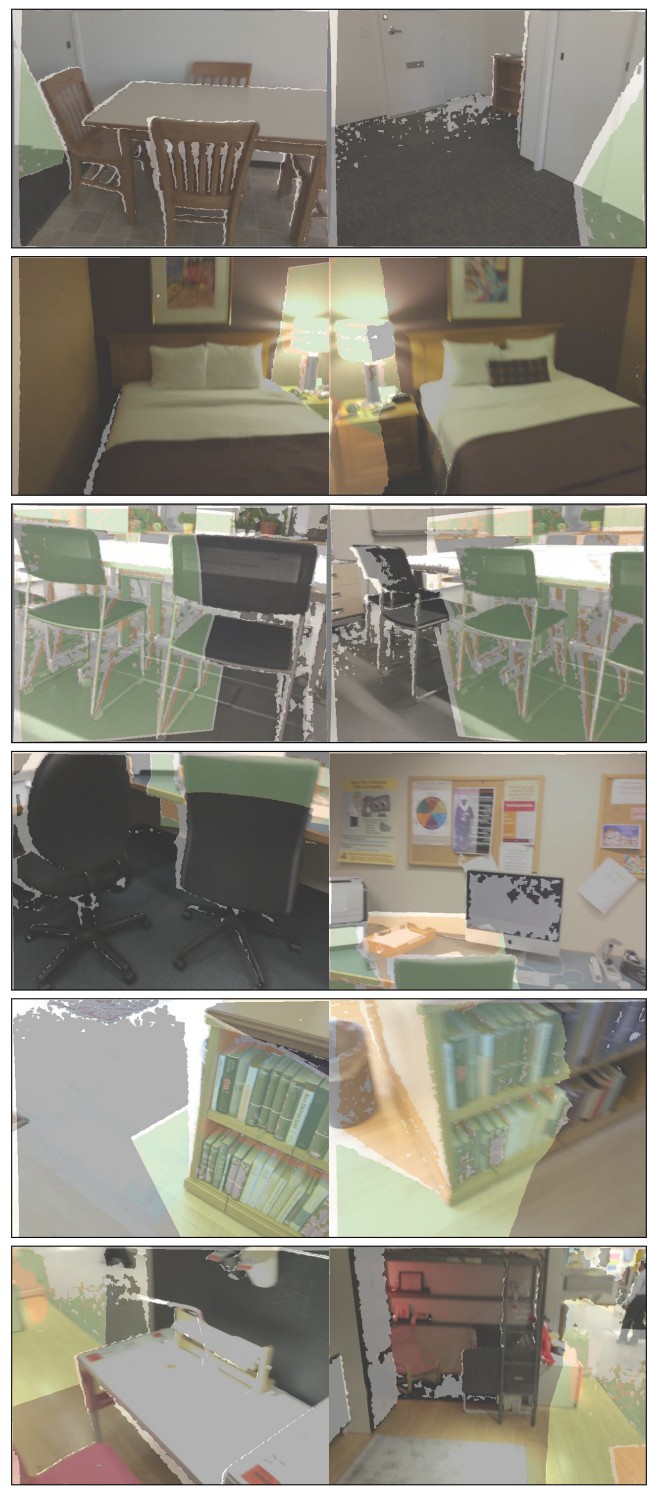

Figure 11: Covisibility annotation examples from Cub3 for ScanNet. For each image pair, we show the corresponding covisibility maps with color-coding for covisible, occluded, and outside FOV regions. Note how our annotation process handles varying degrees of overlap and challenging viewpoint changes.

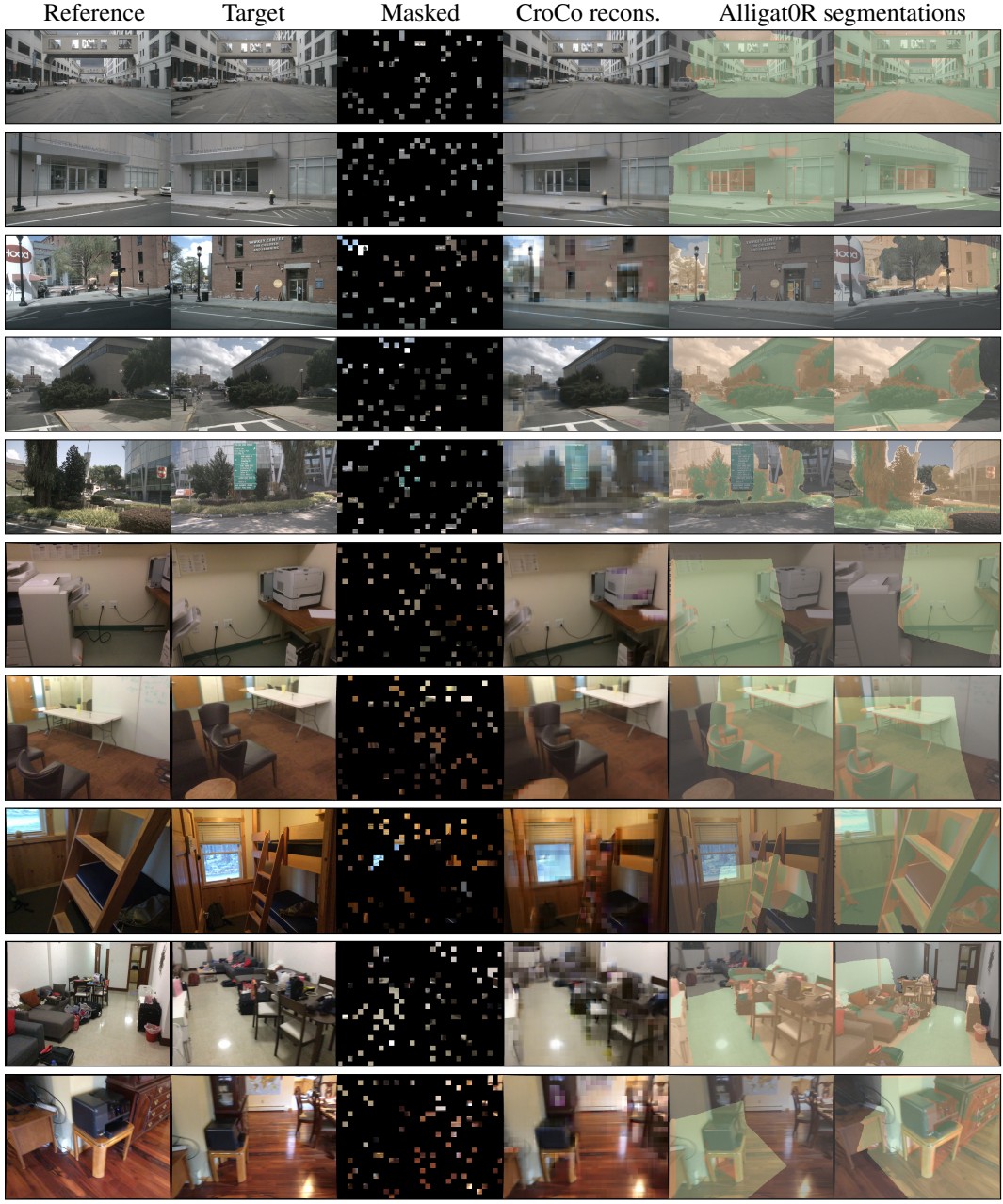

Figure 12: **Qualitative comparison.** CroCo's reconstructions are blurred in masked non-covisible regions, while Alligat0R often correctly identifies covisible, occluded, outside FOV regions across varying degrees of overlap and viewpoint changes.

