# OpenReview forum: "Alligat0R: Pre-Training through Covisibility Segmentation for Relative Camera Pose Regression"
_NeurIPS.cc/2025/Conference — NeurIPS 2025 spotlight_

### Official Review · Reviewer_SujZ · 2025-06-07

**Clarity:** 4
**Significance:** 3
**Originality:** 3
**Rating:** 5
**Confidence:** 4

**Summary:**

The paper defines a pretext task to pretrain a transformer network for geometric computer vision so that the resulting features are suitable for downstream task (e.g. relative pose regression demonstrated in the paper).

Given two images, the pretext task segments each image based on whether the pixel is visible / occluded / oustide of the other image.
The network is made of a self-attention encoder, cross-attentional decoder and a segmentation head.

The network is trained in a supervised manner i.e. each image pairs has an associated "segmentation masks" on two datasets: the indoor ScanNet1500 and the outdoor NuScenes dataset.
These datasets do not readily provide the "segmentation" masks used for training so the masks are generated automatically by leveraging the ground-truth data they provide (e.g. camera poses and depth maps, or camera poses and estimated depth maps).

The relevance of the pretraining is tested on the task of relative pose regression by augmenting the network with a pose regression head.
The network is evaluated on test splits of the ScanNet1500 and NuScenes datasets, which are standard pose estimation datasets.
The experiments compare the result between training the network from scratch, with the proposed pre-training and with another standard pre-training task Croco.
The results show that the proposed pre-training leads to the best performance, which support its relevance.

The contributions of the paper are:
- the definition of a new pre-training task for 3D computer vision tasks
- the pre-training task requires annotation and the paper contributes a training dataset derived from the existing NuScene and ScanNet1500
- The experiments are sound and support the claim of the paper.

**Questions:**

- In the pose regression loss, how are the quaternion parametrized?

- Semantically speaking, occluded and outside the field of view could be considered the same class. Was there any experiment pre-training on two classes only (covisible / not covisible) and was there any insight?

**Ethical Concerns:**

["NO or VERY MINOR ethics concerns only"]

**Final Justification:**

Thank you for the strong rebuttal, it is very much appreciated.
The rebuttal in general has addressed my concerns (data splits and datasets in the evaluation, the use of the pretraining for more tasks than relative pose regression and the framing of the contribution).
There can still be the argument that testing the proposed pretraining on more computer vision tasks would be useful, but the paper is already strong as is, and there is only so many experiments that can fit in a single paper, so the rating is raised to accept.

**Limitations:**

Yes, the authors have adequately addressed the limitations and potential negative societal impact of their work.


### Misc. comments

Strong or unjustified statements:
- L25: "Croco [43, 44] relies on pairs with at least 50% of covisible regions". This is not a strict requirement but covisibility between [40,50]% leads to the best result as shown in Croco's ablation study.
- L31-32: "it [Alligator] aligns more [than Croco] directly with the correspondence reasoning required in downstream binocular vision task". Cross view image completion implies that the network implicitly matches 1 pixel from the 1st image to its matching pixel in the 2nd image so it is also aligned with correspondence-based tasks.
- L36: " providing a more diverse training signal than previous approaches." This is hard to prove as it is hard to quantify a "training signal".
- L90: "Our symmetric design is not only more efficient but also better aligned with downstream binocular vision tasks, which typically process unmasked images." Why is it more efficient? Because there are less parameters to train? Why is it "better aligned with downstream binocular vision tasks"?

Math typo:
- Eq4: $c_{ij}$ should be outside the log

Confusing writing:
- "confirming the results in the original CroCo paper", which results?

**Quality:**

3

**Strengths And Weaknesses:**

### Strengths

- S1 -  The paper proposes a new pre-training task relevant for 3D computer vision task. Previous works have shown how complex yet important pre-training is in 3D vision so the contribution is useful to the community.
- S2 - The paper derives a dataset to enable the proposed pre-training.
- S3 - The paper is extremely well written and pleasant to read.
- S4 - The pretraining and downstream task training are evaluated on both indoor and outdoor datasets.

### Weaknesses

- W1-  Some limitations in the experiments:
  - a) There is no experiment with the off-the-shelf Croco or Croco v2 (trained on real indoor and outdoor images) with and without finetuning. There is no requirement for the Croco pretraining to be on the target dataset and such an experiment would be fairer to Croco as the Croco's author most likely tuned the network to get the best pretraining results.
  - b) The downstream task is evaluated on the same dataset as the pre-training, which prevents appreciating whether the pre-training is generic, which is the case in Croco v1.
  - c) The relevance of the pre-training is evaluated on a single downstream task (relative pose regression). Other tasks would have been relevant although it would have been more computationally demanding: monocular depth estimation, image matching, optical flow.

- W2 - The formulation often argues that proposed pre-training "better" than Croco or addresses its limitation but it would be fairer to say that it is an alternative pretext task. For example:
    - a) Croco's covisibility limitations (i): Croco is trained on image pairs with at least 50% overlap whereas Alligator can train on a pairs with covisibility ranging from [0,100]%. In practice, Croco can also be trained on the full range of covisibility but Croco's authors have run an ablation study showing that the pretraining is best when the covisibility ranges in [40,50]%. Raising the Croco's 50% covisibility requirement as a limitation would be sound only if similar study was run showing that Alligator is insensitive to the covisibility range.
    - b) Croco's covisibility limitations (ii): the 50% covisibility requirement in Croco could be seen as a trade-off for the simpler self-supervision (Alligator requires annotation and at least ground-truth camera poses). Hence the suggestion to frame Alligator as an alternative pre-training method instead of a method better than croco.

---

> ### Author Rebuttal · Authors · 2025-07-31
>
> We thank the reviewer for their constructive feedback and address the main concerns below.
>
> ## **Experimental limitations**
>
> **Off-the-shelf Croco v2 comparison:** We fine-tune the official CroCo v2 on our nuScenes Cub3-all dataset for pose regression and evaluate on RUBIK. We show the results below, keeping only the best Alligat0R model for comparison:
>
> |  |  |  | RUBIK |  |
> |:----------|:-------------:|:-------------:|:-------------:|:-------------:|
> | **Method** | **Backbone** | 5° / 0.5m | 5° / 2m | 10° / 5m |
> | Alligat0R | Frozen | $\underline{24.0}$ | $\underline{55.3}$ | **82.3** |
> | CroCo v2 | Frozen | 5.8 | 15.8 | 34.1 |
> | Alligat0R | Unfrozen | **24.6** | **60.3** | $\underline{81.9}$ |
> | CroCo v2 | Unfrozen | 14.6 | 44.5 | 73.1 |
>
> We can see that our method still outperforms the original CroCo v2 pre-trained on more datasets and fine-tuned for pose regression with our Cub3-all dataset, showcasing the strength of our method.
>
> **Other task evaluation:** To showcase the generalization of our method, we decided to use both our CroCo and Alligat0R models pre-trained on either nuScenes or ScanNet, and evaluate them for the task of zero-shot matching using the correlations from the decoder features on the ETH3D dataset. The metric we use is the Average EndPoint Error (AEPE), computed as the averaged Euclidian distance between estimated and ground-truth flow fields over all valid target pixels (please see *Cross-View Completion Models are Zero-shot Correspondence Estimators*, CVPR 2025). The results are shown in the table below:
>
> |  |  | ETH3D |
> |:----------|:-------------:|:-------------:|
> | **Method** | **Training Data** | **AEPE $\downarrow$** |
> | Alligat0R | nuScenes-all | **43.82** |
> | CroCo | nuScenes-all | 77.61 |
> | Alligat0R | nuScenes-50 | $\underline{58.96}$ |
> | CroCo | nuScenes-50  | 92.98 |
> |  |  |  |
> | Alligat0R | ScanNet-all | **36.07** |
> | CroCo | ScanNet-all  | 56.70 |
> | Alligat0R | ScanNet-50 | $\underline{40.25}$ |
> | CroCo | ScanNet-50 | 86.60 |
> |  |  |  |
> | CroCo v2 | 5 datasets | 51.55 |
>
> As we can see, when trained on the same dataset, Alligat0R always outperforms CroCo for this task, and even beats CroCo v2, showcasing the generalization of our method. An interesting finding is that CroCo trained on all overlaps outperforms its counterpart trained on at least 50% overlap. We will add those results in the camera-ready version.
>
> ## **Framing and claims**
>
> We agree with the reviewer that our method should be positioned as complementary rather than universally superior to CroCo, as it is supervised and targets different assumptions. Our goal is to enable learning from image pairs with arbitrary overlap, and our results confirm that Alligat0R is effective for both pose regression and dense matching under this paradigm. We will modify it.
>
> ## **Specific questions**
>
> **Quarternion parameterization:** We use unit quaternions with L2 normalization, treated as 4D vectors in the homoscedastic loss.
>
> **Two-class experiment:** We actually have this experiment in Table 2 of the Supplementary Material. Results show 3-class formulation provides modest improvements. However, we believe that the model's knowledge about occluded regions might be beneficial for other tasks, especially for pointmap regression. That remains to be tested.
>
> ## **Technical corrections**
>
> We thank the reviewer for their comments and we will implement all corrections and adopt more balanced framing in the camera-ready version while maintaining our core technical contributions.

---

> ### Comment · Reviewer_SujZ · 2025-08-05
> **Rebuttal addresses questions and proposes an alternative formulation of the contribution**
>
> Thank you for the strong rebuttal, it is very much appreciated.
>
> The rebuttal in general has addressed the concerns raised in the review (data splits and datasets in the evaluation, the use of the pretraining for more tasks than relative pose regression and the framing of the contribution).
>
> There can still be the argument that testing the proposed pretraining on more computer vision tasks would be useful, but the paper is already strong as is, and there is only so many experiments that can fit in a single paper, so the rating is raised to accept.

---

### Official Review · Reviewer_uMs9 · 2025-06-20

**Clarity:** 3
**Significance:** 2
**Originality:** 4
**Rating:** 5
**Confidence:** 4

**Summary:**

The authors propose a new multi-view pre-training paradigm for 3D based on covisibility segmentation.
The key contribution is to replace the inpainting objective of CroCo with a cross-entropy loss over segmentation labels (non-covisible, covisible, occluded).
To pre-train the model they collect a dataset of 5M pairs from the nuScenes and ScanNet datasets.
Finally they introduce a relative-pose head that directly predicts a quaternion and metric translation from their backbone features.
The authors show that their approach leads to better relative pose estimates on the nuScenes and ScanNet datasets when compared both to CroCo but also to other SotA methods.

**Questions:**

1. It was not entirely clear to me how the pairs are sampled for ScanNet and nuScenes. Are they sampled uniformly from all scenes except the test scenes in RUBIK and ScanNet1500? If the authors could provide more details here I would be grateful.

If evaluation was conducted on a benchmark other than the training sets, I would likely increase my score to accept (even if the method is not SotA).

**Ethical Concerns:**

["NO or VERY MINOR ethics concerns only"]

**Final Justification:**

The rebuttal clarified my questions regarding the data splits, and results on a benchmark outside of the training domain.
This, as well as the discussion with the other reviews (showing robustness to label noise, etc), is sufficient for me to be confident in this paper, and I have thus increase my score from 4->5.

**Limitations:**

yes

**Paper Formatting Concerns:**

-

**Quality:**

3

**Strengths And Weaknesses:**

**S1 - Elegant objective:** Masked inpainting objectives such as croco have multiple issues. One is that they require masking patches at training time, which is misaligned with how the model will be used in downstream tasks. The second is that if there is no overlap it typically performs poorly. In contrast the proposed objective is very simple, and faces fewer such issues.

**S2 - Strong results:** The results are in general strong, beating out CroCo with a good margin. When finetuned for relative pose they also beat out MASt3R in most cases. The caveat here is that the benchmarks are only conducted on their training datasets, which I go into more detail in W1.


**W1 - Unclear generalization:** All benchmarks in the paper are conducted on the training datasets. While I assume (although I could not actually find this in the paper) that they have excluded these benchmarks from the training data, the fact is that the chosen datasets are not very diverse, and there are likely very similar scenes in the training data as in the test. One could for example investigate map-free relocalization as an alternative benchmark, as the task there is metric pose prediction, to investigate whether the model generalizes. As it stands I think including at least one benchmark on an unseen dataset is vital.

**W2 - Missing/non-standard evaluations:** I was dissapointed to find that there were no comparisons to previous direct relative pose estimation methods, such as Reloc3r. Besides this, the authors have chosen metrics which do not align with the standard AUC metrics in the field, which makes comparisons difficult. I understand that metric relative pose can draw advantage in measuring metric translation errors, instead of angular, but not including any comparisons (except to CroCo) makes this problematic.

---

> ### Author Rebuttal · Authors · 2025-07-31
>
> We thank the reviewer for recognizing the elegance of our objective and the strength of our results. Below, we address the concerns related to generalization and evaluation methodology.
>
> ## **Generalization beyond training datasets**
>
> We would like to clarify that our evaluations are conducted on unseen data, and there is no scene overlap between the training and test sets:
>
> - RUBIK uses test scenes from nuScenes, while our Cub3 dataset uses only the training split of nuScenes, with disjoint scenes.
> - ScanNet1500 is derived from the standard ScanNet test split, whereas Cub3 uses scenes from the ScanNet training split exclusively.
>
> Thus, both RUBIK and ScanNet1500 represent genuine generalization benchmarks in terms of scene disjointness. That said, we acknowledge that the test scenes are from domains that are similar to those seen during training (i.e., same dataset distributions), which may limit out-of-domain generalization.
>
> We also provide non-metric pose regression results in Table 2 of the Supplementary Material, showing strong generalization:
>
> - Alligat0R achieves 20.5 AUC@5 on ScanNet (non-metric)
> - Compared to CroCo, which achieves 13.2 AUC@5
>
> Furthermore, to showcase the generalization of our method, we decided to use both our CroCo and Alligat0R models pre-trained on either nuScenes or ScanNet, and evaluate them for the task of zero-shot matching using the correlations from the decoder features on the ETH3D dataset. The metric we use is the Average EndPoint Error (AEPE), computed as the averaged Euclidian distance between estimated and ground-truth flow fields over all valid target pixels (please see *Cross-View Completion Models are Zero-shot Correspondence Estimators*, CVPR 2025). The results are shown in the table below:
>
> |  |  | ETH3D |
> |:----------|:-------------:|:-------------:|
> | **Method** | **Training Data** | **AEPE $\downarrow$** |
> | Alligat0R | nuScenes-all | **43.82** |
> | CroCo | nuScenes-all | 77.61 |
> | Alligat0R | nuScenes-50 | $\underline{58.96}$ |
> | CroCo | nuScenes-50  | 92.98 |
> |  |  |  |
> | Alligat0R | ScanNet-all | **36.07** |
> | CroCo | ScanNet-all  | 56.70 |
> | Alligat0R | ScanNet-50 | $\underline{40.25}$ |
> | CroCo | ScanNet-50 | 86.60 |
> |  |  |  |
> | CroCo v2 | 5 datasets | 51.55 |
>
> As we can see, when trained on the same dataset, Alligat0R always outperforms CroCo for this task, and even beats CroCo v2, showcasing the generalization of our method. An interesting finding is that CroCo trained on all overlaps outperforms its counterpart trained on at least 50% overlap. We will add those results in the camera-ready version.
>
>
> ## **Data sampling protocol**
>
> We pre-filter all possible pairs from all training scenes with at least 5% overlap for the "all" version of Cub3, and at least 50% for the "50" version. We then sample from all those pre-filtered pairs in order to extract the good number of samples and covisibility masks. The RUBIK benchmark creation is explained from the corresponding paper *RUBIK: A Structured Benchmark for Image Matching across Geometric Challenges*, CVPR 2025 and is extracted from test scenes, and ScanNet1500 benchmark was also created from test scenes of ScanNet and was first introduced in *SuperGlue: Learning Feature Matching with Graph Neural Networks*, CVPR 2020. We will add those details in the Supplementary Material.

---

> > ### Comment · Reviewer_uMs9 · 2025-08-01
> >
> > I thank the authors for the rebuttal.
> > The rebuttal resolves my concern regarding the lack of evaluation on OOD benchmarks with their ETH3D results.

---

### Official Review · Reviewer_MqH9 · 2025-06-29

**Clarity:** 3
**Significance:** 3
**Originality:** 3
**Rating:** 4
**Confidence:** 4

**Summary:**

This paper introduces Alligat0R, a pre-training method for binocular vision that departs from the established cross-view completion paradigm. The authors identify a limitation in prior work like CroCo: the task of reconstructing masked pixels is ill-posed for regions that are not mutually visible (i.e., occluded or outside the field of view). To address this, Alligat0R reframes the pre-training objective as a covisibility segmentation task. The model is trained to predict, for each pixel in one image, whether its corresponding 3D point is covisible, occluded, or outside the field of view in the second image. To support this new pre-training task, the authors have created Cub3, a large-scale dataset of 5 million image pairs with dense covisibility annotations, derived from the nuScenes and ScanNet datasets. The paper demonstrates through extensive experiments that pre-training with Alligat0R leads to significantly better performance on the downstream task of relative camera pose regression compared to a CroCo baseline, especially in challenging scenarios with low image overlap.

**Questions:**

1. Is it possible to use Dust3R/VGGT to provide pseudo depth and depth labels, and hence to generate Covisibility masks? Given that the Covisibility masks for nuScenes were generated by monocular depth estimation methods such as DepthAnything, the quality of Dust3R/VGGT should be sufficient for this case. (Dynamic pixels may be a potential problem, though)

2. Do the authors have any other plan or idea to simplify the generation of Covisibility masks?


The main concerns is about Point 1 in Weaknesses, i.e., pre-training data requirements. The reviewer would be happy to change the rating if the authors could address these issues.

**Ethical Concerns:**

["NO or VERY MINOR ethics concerns only"]

**Final Justification:**

Most of my concerns have been addressed in the rebuttal and hence I raised my rating. I recommend that the authors include these explanations in the final version of the paper

**Limitations:**

Yes

**Quality:**

3

**Strengths And Weaknesses:**

Strengths:

1. The core idea is simple yet seems powerful. The paper correctly identifies a weakness in the cross-view completion pre-training task (e.g., CroCo), its inability to handle non-covisible regions gracefully. The proposed solution of replacing an ill-posed reconstruction task with a well-defined, multi-class segmentation task directly addresses the identified problem.

2. The results are compelling. The head-to-head comparison with CroCo under identical conditions (Table 1) shows an impressive performance gain for Alligat0R.


Weaknesses:

1. The primary weakness of the method lies in its pre-training data requirements. Unlike self-supervised methods that can learn from raw image pairs, Alligat0R requires dense covisibility labels. Generating these labels necessitates access to ground-truth or high-quality estimated 3D information (camera poses and depth maps). Although the authors are transparent about their complex data generation pipeline for nuScenes, this dependency on 3D supervision limits the scalability of the pre-training to datasets where such geometric information is available, and it moves the method away from a purely self-supervised paradigm.

A potential counter-point here is, Croco also requires covisibility in order to filter out image pairs with small overlap. However, it is worth noting that even without filtering, Croco's paradigm still works.


2. The paper convincingly demonstrates the benefits of Alligat0R for relative pose regression. However, as mentioned in the limitation of this paper, the broader claim of pre-training for "binocular vision" would be strengthened by evaluating on other fundamental tasks like stereo matching or optical flow. This is important as Croco has shown the results on these tasks. It remains a question whether the features learned for covisibility segmentation are as effective for these dense correspondence tasks as they are for global pose estimation.

---

> ### Author Rebuttal · Authors · 2025-07-30
>
> We thank the reviewer for their detailed assessment and recognition of our results. Below, we address the primary concern regarding pre-training data requirements, along with other comments.
> ## **Main concern: Pre-training data requirements and scalability**
> We agree with the reviewer that our method requires covisibility labels, which can pose scalability challenges. However, we would like to emphasize that such labels can be obtained in three practical ways:
> 1. Synthetic datasets
> 2. Datasets captured using 3D sensors (e.g., ScanNet)
> 3. Datasets requiring SfM (e.g., nuScenes)
>
> Recent large-scale models such as MASt3R and VGGT leverage combinations of all three types. For example, VGGT training data consists of 59% synthetic, 24% 3D-captured, and only 18% SfM-derived data (see VGGT slides, CVPR 2025).
>
> We acknowledge that SfM pipelines like COLMAP can be computationally intensive and require significant engineering, which we agree departs from the principles of pure self-supervision. However, this limitation can be mitigated through:
>
> - Synthetic and 3D-captured datasets, where covisibility labels are trivial to extract. Indeed, we also trained Alligat0R on ScanNet, which uses 3D sensors, and obtained strong results.
> - Reducing SfM cost via modern alternatives, such as VGGT (as suggested by the reviewer). COLMAP takes ~20 minutes for a 200-image scene, while VGGT can generate camera poses and dense depth maps in ~9 seconds for the same data, a 130× speedup. Moreover, VGGT offers potential quality gains by leveraging context from all frames, as opposed to our current pipeline where depth prediction is done independently per image and then aligned. We believe this trend will continue to improve with advances like VGGT-Long, making the creation of large covisibility-annotated datasets increasingly feasible.
>
> ## **Other plans to simplify covisibility mask generation?**
>
> One potential direction is to use dense matching models that would output two possible labels: covisible or not. From Table 1 in the Supplementary Material, we show that having only two classes works well for the task of pose regression as the model has to be aware of the covisible zones in order to match them and make a prediction. However, for other dense tasks such as optical flow or pointmap regression for example, we believe that the "occluded" class gives a stronger 3D inductive bias that would help solving those particular tasks.
>
> ## **Broader task evaluation acknowledgment**
>
> We agree that framing this work as pre-training for "binocular vision" is important.  To showcase the generalization of our method, we decided to use both our CroCo and Alligat0R models pre-trained on either nuScenes or ScanNet, and evaluate them for the task of zero-shot matching using the correlations from the decoder features on the ETH3D dataset. The metric we use is the Average EndPoint Error (AEPE), computed as the averaged Euclidian distance between estimated and ground-truth flow fields over all valid target pixels (please see *Cross-View Completion Models are Zero-shot Correspondence Estimators*, CVPR 2025). The results are shown in the table below:
>
> |  |  | ETH3D |
> |:----------|:-------------:|:-------------:|
> | **Method** | **Training Data** | **AEPE $\downarrow$** |
> | Alligat0R | nuScenes-all | **43.82** |
> | CroCo | nuScenes-all | 77.61 |
> | Alligat0R | nuScenes-50 | $\underline{58.96}$ |
> | CroCo | nuScenes-50  | 92.98 |
> |  |  |  |
> | Alligat0R | ScanNet-all | **36.07** |
> | CroCo | ScanNet-all  | 56.70 |
> | Alligat0R | ScanNet-50 | $\underline{40.25}$ |
> | CroCo | ScanNet-50 | 86.60 |
> |  |  |  |
> | CroCo v2 | 5 datasets | 51.55 |
>
> As we can see, when trained on the same dataset, Alligat0R always outperforms CroCo for this task, and even beats CroCo v2, showcasing the generalization of our method. An interesting finding is that CroCo trained on all overlaps outperforms its counterpart trained on at least 50% overlap. We will add those results in the camera-ready version.

---

> > ### Comment · Reviewer_MqH9 · 2025-08-05
> >
> > It is a bit weird that my updated final justification seems not visible to others, so I pasted it here:
> >
> >
> > "Most of my concerns have been addressed in the rebuttal and hence I raised my rating. I recommend that the authors include these explanations in the final version of the paper"

---

### Official Review · Reviewer_izzw · 2025-07-02

**Clarity:** 3
**Significance:** 3
**Originality:** 3
**Rating:** 5
**Confidence:** 4

**Summary:**

The paper presents a pre-training scheme for 3D reconstruction and pose regression. The proposed scheme is based on covisibility segmentation where each pixel in one image is predicted as being covisible, occluded or outside the field of view in the second image. This is in contrast to an existing pretraining approach termed CroCro that is based on cross-view completion of image regions in a pair of images. The paper also presents a 5M image-pair dataset with dense covisibility annotations for training and validation of the proposed scheme.

**Questions:**

How sensitive is the proposed scheme to errors in covisibility segmentation?

**Ethical Concerns:**

["NO or VERY MINOR ethics concerns only"]

**Final Justification:**

The authors have addressed my concerns regarding the sensitivity analysis in their rebuttal. Hence I retain my original decision to accept the paper (rating 5).

**Limitations:**

In addition to the limitations mentioned by the authors, the dependence/sensitivity of the proposed technique on/to errors in segmentation should be discussed.

**Paper Formatting Concerns:**

None.

**Quality:**

3

**Strengths And Weaknesses:**

Strengths: The proposed scheme is conceptually simple and intuitive. Unlike the CroCro scheme it does not require a high degree of overlap between the image pairs which is needed for cross-view completion. The proposed scheme can be trained on both, covisible and non-covisible regions in the image pairs unlike CroCro that requires training on regions that are covisible in both images in the image pair. Experimental results show that he proposed scheme performs better than CroCro on most cases in terms of pose regression accuracy and generalizability. The proposed scheme also shows higher pose regression accuracy compared to techniques that do not employ pre-training.

Weaknesses: The proposed scheme would be expected to be sensitive to covisibility segmentation error. This point needs to be discussed accompanied by a formal sensitivity analysis. The notations in the cross-entropy loss function formulation (page 4, line 112) are confusing and need to be clarified/corrected.

---

> ### Author Rebuttal · Authors · 2025-07-29
>
> We thank the reviewer for their thorough and constructive feedback. We appreciate the recognition of our method's conceptual simplicity and intuitive design. Below, we address the specific concerns raised.
>
> ## **Sensitivity to covisibility segmentation errors**
>
> We agree that a proper sensitivity analysis is important. To address this, we are currently training an additional instance of Alligat0R on ScanNet, a dataset with high-quality covisibility labels, while injecting label noise by randomly altering labels with a 20% probability.
>
> Unfortunately, the training job is still running on our supercomputing cluster — due to heavy usage during the NeurIPS rebuttal period, it is progressing slower than usual (the full run takes 20 hours for pre-training, and about 5 hours for fine-tuning). We aim to complete the experiment and populate the table below early during the discussion phase:
>
> |  |  |  | ScanNet1500 | |
> |:----------|:-------------:|:-------------:|:-------------:|:-------------:|
> | **Method** | **Noise** |  10° / 0.25m | 10° / 0.5m | 10° / 1m |
> | Alligat0R | 0% | **85.5** | **92.5** | **95.1** |
> | Alligat0R | 20% |  |  |  |
> | CroCo | NA | 75.7 | 87.4 | 91.5 |
>
> We expect this experiment to demonstrate Alligat0R's robustness to segmentation noise, for the following reasons:
>
> - As discussed in lines 165–171 (Section 4.1) and illustrated in Figure 3 of the main paper and Figure 4 in the Supplementary Material, the model trained on nuScenes already operates with noisy covisibility masks, yet achieves state-of-the-art performance on RUBIK, significantly outperforming CroCo.
> - Additionally, Table 1 in the Supplementary Material shows that collapsing our three covisibility classes into two (i.e., merging categories) has minimal performance impact, indicating robustness to label ambiguity and class boundary errors.
>
>
> ## **Cross-entropy loss notation clarification**
>
> We agree that the current notation is unclear. We will revise and clarify it in the camera-ready version.

---

> > ### Comment · Area_Chair_igWi · 2025-08-03
> >
> > Reviewer izzw, did the rebuttal address your concerns? Do you have any further questions or comments for the authors?

---

> > ### Author Response · Authors · 2025-08-03
> > **Update on sensitivity analysis**
> >
> > We have now completed the sensitivity analysis experiment. The results demonstrate that Alligat0R is indeed robust to label noise:
> >
> > | | | | ScanNet1500 | |
> > |:---|:---:|:---:|:---:|:---:|
> > | **Method** | **Noise** | 10° / 0.25m | 10° / 0.5m | 10° / 1m |
> > | Alligat0R | 0% | 85.5 | 92.5 | 95.1 |
> > | Alligat0R | 20% | **87.7** | **94.9** | **95.9** |
> > | CroCo | NA | 75.7 | 87.4 | 91.5 |
> >
> > Adding 20% random label noise during pre-training actually improves performance across all thresholds (+2.2%, +2.4%, +0.8% respectively). This (somewhat counterintuitive) result suggests that label noise acts as a form of regularization similar to label smoothing, encouraging learning of more generalizable features. Importantly, this finding indicates that our approach remains robust even when covisibility annotations are imperfect.

---

> > > ### Comment · Reviewer_izzw · 2025-08-06
> > > **Sensitivity Analysis to Segmentation Errors**
> > >
> > > Thank you to the authors for performing the sensitivity analysis to segmentation errors. The observation that noisy labels act as a regularizer is very interesting. Experiments with different noise levels should enable the authors to test the limits of this regularization property. I will retain my original decision to accept the paper.

---

### Decision · Program_Chairs · 2025-09-17

**Decision:**

Accept (spotlight)

**Comment:**

This paper proposes Alligat0R, a pre-training scheme for binocular vision that enhances downstream applications, such as camera pose regression. The method's core idea is to predict a covisibility segmentation map for a second image, classifying pixels as covisible, occluded, or outside the field of view.

During the review process, the reviewers appreciated the simplicity and effectiveness of the proposed pre-training framework, noting the improved results and the high quality of the writing.
However, the reviewers also raised several concerns, including:
1. The method's sensitivity to errors in the training data.
2. The necessity of ground truth camera poses and depth maps to generate training data.
3. The evaluation focused on global relative pose estimation rather than local computer vision tasks like stereo matching.
4. The evaluation was conducted on test splits from the training domain rather than on out-of-domain data.

During the rebuttal and discussions, most of the concerns have been solved. The authors provided a strong rebuttal with new experiments, showing robustness to label noise and successfully demonstrating generalization to a new task and an out-of-domain dataset. Consequently, all the reviewers are positive about the paper with final scores: 5,5,5,4.
Therefore, I believe that this paper should be accepted to NeurIPS.